# A New, Practical Animal Welfare Assessment for Dairy Farmers

**DOI:** 10.3390/ani11030881

**Published:** 2021-03-19

**Authors:** Frank J. C. M. van Eerdenburg, Alice M. Di Giacinto, Jan Hulsen, Bert Snel, J. Arjan Stegeman

**Affiliations:** 1Department of Population Health Sciences, Faculty of Veterinary Medicine, Utrecht University, 3584 CL Utrecht, The Netherlands; j.a.stegeman@uu.nl; 2Department of Animal Sciences, Adaptation Physiology Group, Wageningen University, 6708 PB Wageningen, The Netherlands; alicedivi@gmail.com; 3Vetvice/CowSignals, 4614 PC Bergen op Zoom, The Netherlands; Hulsen@vetvice.nl; 4DLV Advies, 7418 EV Deventer, The Netherlands; Bert.sneladvies@gmail.com

**Keywords:** animal welfare, dairy cattle, Welfare Quality^®^, water supply, integument alterations, economic return

## Abstract

**Simple Summary:**

To determine the level of welfare on a dairy farm is a complex task. There is no protocol available that can serve as a ‘gold standard’. The Welfare Quality protocol is the most extensive one, but it takes about a full day to perform. We, therefore, examined if it would be possible to replace the time-consuming parts, like lengthy behavioural observations, with simple measurements in the environment. This resulted in a new Welfare Monitor that can be executed in 1.5 h on a farm with 100 cows. Welfare assessment was appreciated by the farmers, and they responded to advice for improvements on their farm. Besides promoting better welfare, this approach also led to a better financial result for the farms.

**Abstract:**

The Welfare Quality^®^ assessment protocol (WQ) is the most extensive way to measure animal welfare. This study was set up to determine if resource-based welfare indicators, that are easier and faster to measure, could replace the more time consuming, animal-based measurements of the WQ. The WQ was applied on 60 dairy farms in the Netherlands, with good, moderate and poor welfare. The WQ protocol classified most farms (87%) as ‘acceptable’. Several of the animal-based measures of WQ correlated well with measures in the environment. Using these correlations, an alternative welfare assessment protocol (new Welfare Monitor) was designed, which takes approximately 1.5 h for a farm with 100 dairy cows. Because the opinion of farmers about welfare assessment is important if one wants to improve conditions for the cows at a farm, another objective of this study was to evaluate the usefulness of the new Welfare Monitor for the farmer. Over two years, the farms were visited, and advice was given to improve the conditions at the farm. After the first welfare assessment and advice, farmers improved the conditions for their cows substantially. Farms where the category score had increased made more improvements on average than those that did not upgrade.

## 1. Introduction

Although there is no specific EU directive for dairy cows, a recent report by the EU Parliament’s Directorate-General for Internal Policies stated that dairy cow welfare might be considered to be the second greatest animal welfare problem in the EU [1,2]. The first step towards the improvement of cow welfare on farms is accurate and frequent welfare assessment. This requires a reliable assessment protocol that can be executed in 1–2 h. However, a ‘gold standard’ for welfare assessment is still lacking. Several protocols have been developed that measure animal welfare at dairy farms, of which the Welfare Quality assessment protocol^®^ [3] (WQ) is the most extensive one, and it uses mainly animal-based measures (ABM). These are parameters that are measured directly on/from the animals, like skin lesions or behaviour, and not in the environment (resource-based parameters (RBM)). In the WQ, 33 measures are taken on a farm and are integrated into 12 criteria. These 12 criteria are then further grouped into four principles: Good feeding, Good Housing, Good Health, and Appropriate Behaviour. Finally, an end classification is calculated, and this can be Excellent, Enhanced, Acceptable, or Not Classified [3]. However, the execution of the extensive WQ protocol [3] is time-consuming (almost a full day is needed), which has hampered its implementation as a routine welfare check on a dairy farm [4,5]. Furthermore, the relative contribution of certain measures/criteria to the end classification of the WQ is disputed [5,6,7,8]. For example, in the study of Heath et al. [5], it appeared that they could classify the final outcome of the WQ protocol [3] correctly in 88% of the farms, with the result only for “Absence of prolonged thirst”. Furthermore, the discriminative capacity of the WQ is also disputed because most farms are classified as ‘acceptable’ in the studies of De Vries et al. [6], Heath et al. [5], De Graaf et al. (ILVO, Gent, Belgium, personal communication) in Belgium, who classified 94 out of 111 farms as acceptable (none not classified or excellent) and Toma et al. [9] who found nine farms with enhanced, 25 acceptable and one not classified in a study in Scotland. For regular use, the execution time of the protocol needs to be shorter. In the present study, several RBM, which can be measured in a shorter period, were compared with the ABM of the WQ. By replacing a number of ABM with RBM, we were able to construct a Welfare Monitor (WM) that could be executed in 1.5 h.

So far, little attention has been paid to the farmers’ role and opinion on welfare assessment [10,11]. This is remarkable since the farmers’ work and attitude have a direct impact on the well-being of animals [12,13]. Several studies have shown that farmers consider that animal well-being is important for numerous reasons. Farmers show an honest interest in the intrinsic value of the animals [10] and have expressed that working with healthy animals gives greater job satisfaction [14]. Furthermore, animal welfare is positively related to production [15,16]. Therefore, a fast protocol that can be implemented in routine management checks is desired [4,5].

Studies in Austria, Germany, Italy, and Denmark show that farmers are intrinsically motivated and actually willing to make changes in their management according to welfare assessment techniques, but they have concerns about complexity and profitability [11,17]. The researchers in these studies suggested that it would be useful and increase acceptance by the farmers if these systems were correlated with production results or economic incentives. This would be an opportunity to learn from mistakes and successes in order to provide cost-effective interventions to improve animal welfare [18]. Acceptance of the protocol used by the farmers is essential to get it implemented in their management and change their behaviour [19]. Therefore, another objective of the present study was to evaluate the usefulness of welfare assessment for the farmer. Based on the results of de Vries et al. [6], we expected that farms with higher welfare scores would have higher annual milk returns. Because the end result of the WQ and also the Welfare Monitor have only four categories, and the farms scored initially mainly in only two of them, an attempt was made to divide these categories into subcategories; as was done by Tuyttens, et al. [20] for poultry, in order to determine if this would give a more detailed discrimination between the farms. In this way, farmers could increase in category score with less difficulty, in the expectation that this will encourage them to improve animal welfare at their farms.

## 2. Materials and Methods

The WQ protocol [3] was applied to 60 dairy farms in the Netherlands. Four large veterinary practices, spread over the Netherlands in order to avoid possible regional effects, were asked to make a list of their dairy farmer clients. All the dairy cattle veterinarians of each practice classified each farm as good, average or bad, based on the availability of good quality food and water, quality of housing, health and behaviour. This classification was based on the impression of all the dairy cattle veterinarians of each practice, in consensus with Botreau et al. [21]. These were large veterinary practices, each with more than 5 dairy cattle veterinarians, so opinion was not subjective or individual. Out of the lists, randomly, in each of the 4 veterinary practices, 5 good, 5 average and 5 bad farms were selected. This was not used as a ‘gold standard’ nor a representative sample of the Dutch dairy farms, but just to get a diverse quality of farms in order to evaluate the assessment protocol over the full range of animal welfare status. The selected farmers were asked if they would be willing to participate in the project and, if not, the next farmer on the list was addressed. This occurred twice. During the project, one farm dropped out due to the fact that they started a substantial rebuilding of the barn.

In order to execute the WQ protocol in the right way, a course is required [3]. Of each practice, at least one veterinarian was trained to execute the WQ protocol during a three-day course provided by the Welfare Quality consortium. The observers did not assess farms that they regularly visit and advise because of possible bias. The observers assessed farms in the area of another veterinary practice.

After the initial assessment, it became evident that the WQ protocol had a low discriminative capacity because most farms (87%) were rated as acceptable. In Figures 3–5, the results are presented of the extremely lean cows, severely lame cows and the number of cows with skin lesions of the 60 farms to give an indication of the level of problems. This was confirmed by De Vries et al. [6], Heath et al. [5], Toma et al. [9], and data from Tuyttens et al. (unpublished results) who did a survey in Belgium and 94 farms were rated as acceptable versus 17 enhanced (none not classified or excellent). Therefore, we made 3 modifications to the original WQ protocol [3] in order to increase the discriminative capacity [1]. These modifications were included in the new Welfare Monitor (see below). Because most of the calculations of the new Welfare Monitor are the same as in the WQ protocol, references are given to the WQ protocol [3] when applicable.

### 2.1. New Welfare Monitor

Since the new Welfare Monitor (WM) was designed based on the WQ protocol [3], it also had four principles: good feeding, good housing, good health, and appropriate behaviour (Table 1). A full description (including the calculations) of the new WM is available as Appendix A. In order to reduce the time needed for an assessment, the number of animals for clinical scoring was reduced. After the initial scoring, according to the WQ protocol [3], the data from animals were removed from the dataset in a systematical way. First, every fourth animal was removed (25% reduction). This procedure was repeated with every third animal (33% reduction) and finally, with every second animal (50% reduction). The outcome of the clinical scoring was compared with the scoring of 100%.

#### 2.1.1. Principle 1: Good Feeding

Assessment of the feeding status was identical to the WQ protocol [3]. However, a weighted score for the cleanliness of the drinkers was used. A clean drinker scored 1, a partially dirty 2, and a dirty one 3 points. After giving the score for the rest of the drinking-related parameters measured, the total was divided by the average score for the cleanliness (see Figure 1). This number was then used in the calculations according to the WQ protocol [3].

#### 2.1.2. Principle 2: Good Housing

The measures used to compute the principle of good housing are as follows:

The width (distance between the dividers) and the diagonal of the freestall (distance of the neck rail to the curb) are used in the new Welfare Monitor, and the ‘barn environment’ as well as the softness of the bedding were also included in the new protocol. Furthermore, the way the cleanliness of the cows was measured in the WQ protocol [3] is also rather time-consuming and complex, and this was, therefore, done in a different way. The weight of the parameters and calculations are the same as in the WQ protocol [3]. This resulted in the following measurements and calculations:

##### Dimensions of the Cubicles

-If Diagonal ≤ 185 cm = 9 points; else if 185 cm < Diagonal < 195 cm = 4 points; else = 0 points.-If Width ≤ 110 cm = 9 points; else if 110 cm < Width < 120 cm = 4 points; else = 0 points (both measured as space between the tubing).-If % Lying outside the stall ≥ 2% = 9 points; else if 2% > % Lying outside the stall ≥ 0% = 4 points; else = 0 points

These 3 scores needed to be multiplied by 3 and summed to calculate A.

##### Cleanliness of the Animals (Hygiene)

The size of the dirty parts of the skin of the cows is measured during the clinical inspection (WQ). The number of points belonging to the percentage of cows is presented in Table 2. The sum of the points is the score for hygiene (H). If 15 > H ≥ 9 then B = 0 points; if 9 > H ≥ 7 then B = 4 points; else B = 9 points.

Clinical scoring for dirtiness of the skin. The percentage of cows having each category of dirty patch size was calculated and marked with 1–5 points. These were summed. Example: 1.5% of the cows had a dirty patch size 25 × 25–50 × 50 cm; 0.6% had a dirty patch 50 × 50 cm–one half hind quarter, and 0.3% was dirty > one half hind quarter. This would result in 3 + 2 + 2 = 7 points. The score for hygiene H = 7 and B = 4. For the hygiene of the cows, the weight was one-third of the rest of the factors in this calculation. This was similar to the WQ protocol [3].

The softness of the bedding was measured with the knee test [22] and could be classified as Good (soft), Moderately good, or Insufficient (hard). For this test, one drops from a standing position on his/her knees onto the bedding without touching anything. The level of pain experienced is the outcome. When it is Good: C = 0; Moderately good: C = 4; Insufficient: C = 9.

The barn environment is measured in three parameters: light, ventilation and the presence of a mechanical brush. Each parameter can be good, partly good or insufficient.

Light:Good—everywhere in the barn; it is easy to read a newspaper;
Partly—only at the feeding fence and some other places;
Insufficient—(almost) nowhere in the barn.Ventilation:Good—air in the barn smells fresh and ample options for ventilation;
Partly—the air smells not-so-fresh, and there are not many ventilation options;
Insufficient—air is dirty, and there are few options for ventilation.Mechanical brush: Present or not.

The flow chart for the calculation of the score for the barn environment (D) is presented in Figure 2.

First, the light was checked: Good—everywhere in the barn, it is easy to read a newspaper; Partly—only at the feeding fence and some other places; Insufficient—(almost) nowhere in the barn. Second, the ventilation was checked: Good—air in the barn smells fresh and ample options for ventilation; Partly—the air smells not-so-fresh, and there are not many ventilation options; Insufficient—air is dirty and few options for ventilation. Finally, it was checked if there is a mechanical brush present or not. Finally, the index for comfort around resting (P) was calculated.
P = 100 − 100 * (A + B + C + D)/108

The sum was divided by 108 because of the theoretical maximum of the sum. The score is computed according to the WQ protocol [3].

#### 2.1.3. Principle 3: Good Health

This was identical to the WQ protocol [3] with one modification, which was in the ‘absence of injuries’. In the new Welfare Monitor, the hairless patches (HPs) and lesions/swellings were assessed and counted as in the WQ protocol [3]. However, the average number of HPs, lesions and swellings per cow in the group was used in the calculations. Because a lesion or swelling is a more severe impairment for the welfare of the cow, it received more weight in the calculations, similar to the WQ protocol [3].

The Index for integument alterations was calculated as:$$I=100-\frac{\left(2\mathrm{HP}+5\left(\mathrm{lesions}+\mathrm{swellings}\right)\right)\times 10}{5}$$

If I ≤ 65 the score becomes: (0.43 × I) + (0.0065 × I^2^) + (0.00013 × I^3^)

If I > 65 the score becomes: 29.9 − (0.94 × I) + (0.015 × I^2^) + (0.00002 × I^3^)

Where HP is the average number of HP’s per cow and lesions + swellings are also the average number of lesions and swellings per cow. This index was then used instead of the one for integument alterations from the original WQ protocol [2] in the calculations.

#### 2.1.4. Principle 4: Good Behaviour

The avoidance distance at the feeding fence (ADF) was measured according to the WQ protocol [2]. In the result of this test, the cows were grouped into 4 groups: 0 cm (can be touched); 0–50 cm; 50–100 cm; >100 cm. The correlation (r^2^) between social behaviour of the WQ protocol [2] with the ADF group 3 was 0.833 (*p* = 0.11). Therefore, this result was used in the new protocol to replace the time-consuming watching and sometimes difficult interpretation for social behaviours. The formula is 100 − % of cows in ADF group 3.

This results in the calculation for the principle of Good Behaviour:
B_1_ = Expression of social behaviours: 100 − % of cows in ADF group 3.B_2_ = Expression of other behaviours as in the WQ protocol [3] (access to an area outdoors)B_3_ = Index for good human animal relationship: ADF + calculations as in the WQ protocol [3]
$$\begin{array}{l} \mathrm{Principle}\;\mathrm{for}\;\mathrm{Good}\;\mathrm{Behaviour}=\{\begin{array}{c} B_{1}+\;\left(B_{2}<B_{1}\right)\mu_{23}+\;\left(B_{3}<B_{2}\right)\mu_{3}\\ B_{1}+\;\left(B_{3}<B_{1}\right)\mu_{23}+\;\left(B_{2}<B_{3}\right)\mu_{2}\\ B_{2}+\;\left(B_{1}<B_{2}\right)\mu_{13}+\;\left(B_{3}<B_{1}\right)\mu_{3}\\ B_{2}+\;\left(B_{3}<B_{2}\right)\mu_{13}+\;\left(B_{1}<B_{3}\right)\mu_{1}\\ B_{3}+\;\left(B_{1}<B_{3}\right)\mu_{12}+\;\left(B_{2}<B_{1}\right)\mu_{2}\\ B_{3}+\;\left(B_{2}<B_{3}\right)\mu_{12}+\;\left(B_{1}<B_{2}\right)\mu_{1} \end{array}\;\begin{array}{c} \mathrm{if}\;B_{1}<B_{2}<B_{3}\\ \mathrm{if}\;B_{1}<B_{3}<B_{2}\\ \mathrm{if}\;B_{2}<B_{1}<B_{3}\\ \mathrm{if}\;B_{2}<B_{3}<B_{1}\\ \mathrm{if}\;B_{3}<B_{1}<B_{2}\\ \mathrm{if}\;B_{3}<B_{2}<B_{1} \end{array}\\ \begin{array}{c} {µ}_{1}=\;0.20\\ {µ}_{2}=\;0.14\\ {µ}_{3}=\;0.24 \end{array}\;\begin{array}{c} \mu_{12}=\;0.24\\ \mu_{13}=\;0.24\\ \mu_{23}=\;0.30 \end{array} \end{array}$$

### 2.2. Farmers Opinion

The farms were visited twice a year (April and November) for 2.5 years, from 2013 to 2015. After each assessment, the farmers not only received advice for improvements but also sessions were organized with the 15 farmers of each participating veterinary practice to discuss the results and advice. At the end of the project, a survey was sent to the farmers (See Appendix A Survey farmers welfare monitor data). They were asked which of the recommendations they had implemented. Furthermore, the farmers also gave their opinion on whether or not they agreed with the scores for the welfare assessment their farms had received during the whole project. The annual milk returns (AMR) of the farms for April 2015 were available. The AMR is the economic profit of a yearly milk production per cow. To calculate it, the kg of fat and protein in the milk were taken into account [23]. Since the AMR was calculated every 3 weeks, we used the average between the welfare scores of November 2014 and April 2015 to make the comparisons with the AMR at the farm level.

#### 2.2.1. Statistical Analysis

After a descriptive analysis of the data (means, SD), ordinal logistic regression (OLR) was used to test if the improvement in score increased the probability of having a better opinion of the scores. Then the analysis of variance was performed (one-way ANOVA) to test if the farmers that had a higher improvement of their total sum score were the ones who agreed with their score. To determine which level of agreement differed from one another, the protected Fisher’s Least Significant Difference (LSD) test was used. This method was also used to test if there was a difference in the number of improvements made among the farmers that agreed and those that did not agree with their scores.

#### 2.2.2. Improvements

To analyze if the total number of improvements were related to a change in the welfare category of the WM, a one-way ANOVA was utilized. OLR was used to determine if there were particular types of improvements that increased the probability of achieving a higher category score. Then we employed simple and multiple linear regression to assess if the total number, or a particular type of improvement, influenced the progress in the total sum score. The same process was followed for each principle score.

#### 2.2.3. Categorizing System

After the first assessment and advice, a substantial improvement was achieved in welfare status at most farms due to improvements in management and housing. After this initial improvement, the changes in the category were minimal. Therefore, the categories were divided so that there were more (sub)categories and that the difference between category thresholds was smaller. In this way, farmers could increase in category score with less difficulty, in the expectation that this would encourage them to continue to improve the animal welfare at their farms. A comparison between the current and the new categorizing system is presented in Table 3.

Farms were re-categorized for the evaluations of November 2013 and April 2015, and then the same tests performed as before for the category score (one-way ANOVA and ORL) were used to compare the new categories to the improvements made.

#### 2.2.4. Economic Efficiency

To check if the AMR changed according to the farms’ welfare category, an independent sample *t*-test was conducted. To test to what extent the total sum and principle scores influenced the AMR, we used multiple linear regression.

All the statistical analysis was performed in SPSS 22 (IBM SPSS Inc., Chicago, IL, USA).

## 3. Results and Discussion

Three farms (5%) scored Not Classified, 52 (86.7%) Acceptable, and five (8.3%) Enhanced; no farm received a score Excellent under the original WQ protocol [3]. Since the farms were selected as having bad, average or good welfare in equal numbers, this was not expected. Because there is no ‘gold standard’ for animal welfare assessment available, some degree of subjectivity is inevitable when weighing different measures [24]. So it could be that the farms were not selected in an appropriate way. However, analysis of the measurements of the farms showed that there were indeed substantial differences between welfare determining parameters on farms (Table 1; Figure 3, Figure 4 and Figure 5). The way of selecting the farms was comparable with Botreau et al. [21], who used the ‘general impression’ of the observers of the farms in their study to compare the procedures that could form the basis of the calculations of aggregation of the measures in the WQ protocol. In the end, the way of computing, that matched the ‘general impression’ of the observers in the best way, was implemented in the WQ protocol as a final step to categorize the farms [21]. In the present study, not just the general impression of one person was used, but several persons based their opinion on the availability of good quality food, water, quality of housing, health, and behaviour, and this was only to select a wide variety of farms to execute the welfare assessments. The initial classification by the veterinarians was not used in the calculations nor compared with the results of the WM or WQ.

As can be seen in Figure 3, Figure 4 and Figure 5, several farms had a substantial amount of problems, e.g., 18 farms had 10% or more severely lame cows (Figure 4), a disorder with a substantial impact on animal welfare. According to the WQ protocol [3], this can be ‘acceptable’ since only three farms were considered ‘not classified’. The body condition score of the cows was also a problem on a large number of farms (Figure 3). Similar findings have been reported previously. In a study in England and Wales by Heath et al. [5], all the 92 farms they assessed had a result as acceptable (35 farms) or enhanced (57 farms). Data from de Graaf et al. in Belgium (ILVO, Gent, Belgium, personal communication) confirmed this. Out of 111 farms they assessed 94 as acceptable vs. 17 as enhanced (none not classified or excellent). Furthermore, Toma et al. [9] categorized nine farms as enhanced, 25 acceptable and one not classified in a study in Scotland.

The correlation between principle scores and the WQ protocol evidenced that Good feeding, and Appropriate behaviour were the main principles influencing the classification. The other two appeared of minor importance (Table 4). De Vries et al. [6] also reported that a limited number of welfare measures had a strong influence on the WQ classification of dairy herds. This was confirmed in the study of Heath et al. [5], where 88% of the farms could be classified correctly with “absence of prolonged thirst”, a component of the first principle, only. De Graaf et al. [8] also reported that the absence of prolonged thirst and the Qualitative Behaviour Assessment were the most influential measures. Heath et al. [5] suggested that the protocol could be shortened to just 15 min with the same outcome. This is improved in the new Welfare Monitor, where all four principles contributed to the end classification (Table 4).

Replacing animal-based by environment-based measures not only saved time but increased the reliability of the measurements and also provided the farmer with clues to improve the welfare situation at the farm [25].

The number of animals for the clinical inspection could be reduced substantially without changing the outcome of the protocol, as is presented in Table 5. The deviation was <10% even if only 50% of the animals were used for the clinical inspection.

### 3.1. Principle 1: Cleanliness of the Drinkers

If on a farm there was one dirty (or partially dirty) drinker, not all drinkers were clean. The question in the calculation of the WQ protocol is: “Are the drinkers clean?” (WQ p. 95 [3]). This then had to be answered as ‘No’, resulting in a maximum score for the absence of prolonged thirst of 32 out of 100 points. This implies that on a farm with 100 cows with 12 water bowls with sufficient flow and of adequate size, on at least two different locations, the score for the absence of prolonged thirst would be 32 points if one of the drinkers was (partially) dirty and 11 were clean. These are more clean drinkers than minimally required by the WQ protocol [3]. On another farm with 100 cows with seven water bowls with sufficient flow and of adequate size, on at least two different locations, the score for the absence of prolonged thirst would be 60 points if all seven drinkers were clean. This implies that the WQ protocol [3] considers the water supply almost twice as good when there are four clean drinkers less available for the animals. This is, in our opinion, not correct. However, in practice, a farmer cannot clean each drinker several times a day; often, one of the drinkers will be (partially) dirty when the assessor is at the farm. This implies that, in practice, the maximum score for the absence of prolonged thirst would be 32 points. Even when the score for the absence of prolonged hunger was maximal (100 points), the score for the first principle would be 40.16 points. This was not even considered ‘enhanced’ by the WQ protocol [3]. In our new Welfare Monitor, therefore, the weighted score for (mean) cleanliness of the drinkers (see the Materials and Methods section) was introduced. In this way, a single dirty drinker cannot determine the score for the absence of prolonged thirst and thus the score for the first principle.

### 3.2. Principle 2: Good Housing

In the WQ protocol [3], the number of collisions with the dividers of the freestalls was counted, and the average time to lie down was measured during lengthy observation periods. The results of the present study revealed that there were correlations with several dimensions of the freestall. The number of collisions with the dividers correlated with the width of the freestall (r^2^ = 0.63; *p* < 0.03). This seemed logical since in a narrow freestall a cow will touch the dividers more often. The time needed to lie down showed a trend with the diagonal of the freestall (distance of the neck rail to the curb) (r^2^ = 0.24; *p* < 0.06). The diagonal determines the space available to move forward when lying down. These freestall dimensions were, therefore, used in the new protocol. The ‘barn environment’ had a correlation with the principle of good housing (r = 0.43; *p* < 0.01), and the softness of the bedding as measured in the new Welfare Monitor showed a trend (r = 0.23; *p* < 0.08). Both were also included in the new protocol. Furthermore, the way the cleanliness of the cows was measured in the WQ protocol [3] was also rather time-consuming and complex. The correlation for this item between the WQ protocol [3] and the new Welfare Monitor was one (*p* < 0.000).

### 3.3. Principle 3: Good Health

For the criterion ‘Integument Alterations (hairless patches and lesions/swellings)’, the WQ protocol [3] takes into consideration if a cow has one or more HP’s, swellings or lesions. The classification in the WQ protocol [3] is as follows: “Percentage of animals with no integument alteration (no HP, no lesion/swelling). Percentage of animals with mild integument alterations (at least one HP, no lesion/swelling). Percentage of animals with severe integument alterations (at least one lesion/swelling)”. However, the number of these alterations per animal nor the severity is taken into account. A cow with 20 HP was the same in the calculations as one with just one, and a lesion of 20 cm^2^ was the same as one of 3 cm^2^. This did not seem right because it will make a difference in the level of pain experienced by the cow if there were multiple lesions. So in the new Welfare Monitor, the average number of HP/lesions/swellings per cow was used in the calculations.

### 3.4. Principle 4: Appropriate Behaviour

The correlation (r^2^) between social behaviour of the WQ protocol [3] with ADF group 3 was 0.83 (*p* = 0.11). Therefore, this result is used in the new protocol to replace the time-consuming watching, and sometimes difficult interpretation, of social behaviours. 

### 3.5. New Welfare Monitor

So, in short, the new Welfare Monitor is based on the WQ protocol with several substitutions of resource-based measurements for lengthy observations of the herd. Furthermore, several calculations have been modified to increase the discriminative capacity.

The question is, of course: Why create another protocol? To answer this question, the first argument is that the WQ protocol takes too much time to be used as a practical tool [4]. Therefore, the Danish Cattle Federation has developed a protocol that correlates well with the original WQ protocol [3] and takes 2 h to execute [4]. But, as explained before, the WQ has low discriminative power. In Sweden, a protocol has been developed that uses the outcome of measures of all Swedish farms to determine the welfare level of a particular farm [26]. It uses the recorded data of all farms, and if a farm does not score in the 10% worst cases for a measure, it is classified as a farm with good welfare [27]. So if most farms have a bad score for one measure (e.g., % of lame cows), this will be the standard. The WQ [3] does not take into account what most farms score, but what a farm should score, based on what is considered acceptable from expert opinions. We also think that that is the way to go. However, on the basis of all protocols in use lies the wish to improve the welfare status of the dairy cows. Whether a protocol will be successful in achieving this, largely depends on the attitude of the farmers [13]. They prefer a quick and straight forward approach. The assessment protocol described here fulfils these requirements and can be implemented in a routine farm-management check.

### 3.6. Appreciation by the Farmers

Table 6 shows the category, principle score and the total sum score obtained by the farms in the years 2013 and 2015. After the first assessment, a substantial improvement was observed. After the second assessment (Nov 2013), the improvements were not so evident. However, there were no farms ‘not classified’. Table 7 shows the differences between the two years. The recommendations that were made to the farmers are presented in Table 8.

#### 3.6.1. Statistical Analysis

An overview of the results of the statistical analysis with the *p*-values are shown in Table 9.

#### 3.6.2. Opinion of the Farmers

No significant relationship was found between the opinion of the farmers and the progress in the scores (*p* = 0.59) nor with the improvements they made on the farm (*p* = 0.36). In further analysis, a positive relationship was found between the last total sum score (April 2015) and the opinion of the farmers (*p* = 0.038). Although the farmers were enquired on their opinion of the scoring from the two years of the project, they possibly only considered their last scoring or other factors. That could be the reason why opinions and score progress were not related, but the last scores and the opinion was. This sends the message that farmers that have problems with the welfare in their farms might not be aware of it or do not consider certain issues as a problem for animal welfare.

#### 3.6.3. Improvements

During the study, it became clear that the farmers indicated a preference for a quick and straight forward approach. Farms that had increased their category score by one had made on average 7.44 more improvements than those that did not upgrade (*p* = 0.007) and 12 more than those that decreased in category score (*p* = 0.01). There was no significant difference between the farmers that did not improve and those that decreased in category score. The health improvements were the ones that significantly affected the probability (in 1:1.5) of increasing the category score (*p* = 0.014).

Regarding the total sum score, there was a relationship between the total number of improvements made by the farmers and the total score (odds ratio (OR) 1:3.032; 95% confidence interval (CI) 0.971, 9.47, *p* = 0.001). When each type of improvement was tested, it was the housing improvements that showed a significant influence on the total sum score (OR 1:1.91; 95% CI 1.29, 2.81, *p* = 0.015). 

When the improvements were classified into the principles involved, the housing and behaviour principle scores were positively related to their own type of improvements (OR 1:2.47; 95% CI 1.54, 3.98, *p* = 0.001 and OR 1:1.84; 95% CI 1.06, 3.19, *p* = 0.015 respectively). In further analysis, it was found that the housing principle was also negatively affected by feed improvements (OR 1:0.56; 95% CI 0.37, 0.84, *p* = 0.007). When doing stepwise regression, apart from housing and behaviour, the health score also showed a positive connection (OR 1:2.88; 95% CI 0.99, 8.29, *p* = 0.031) with its respective type of improvements; and a positive interaction was found between housing improvements with the feed principle score (OR 1:1.46; 95% CI 1.084, 1.98, *p* = 0.006). The results regarding the welfare categories (good, acceptable, etc.) and the total sum scores (the sum of the four principle scores) led to the conclusion that the improvements most probably ameliorated the overall welfare of the animals in the farms on different levels. Whereby health and housing improvements had a significant individual effect on the category and total sum score, respectively. 

Health is a principle that is very interrelated with others. Animals with poor health tend to have consequential issues like weight loss and behavioural changes. It is obvious that diarrhoea leads to dirtier animals [28]. These issues affect the feed, behaviour and housing scores, respectively. For example, lameness will alter the behaviour and time budgets of cows [29,30], and Fogsgaard et al. [31] found that cows having an *Escherichia coli* mastitis showed changes in behaviour; they spent less time eating and ruminating than healthy animals. Although housing is not as interrelated with the other principles as health, housing improvements can have more direct and immediate effects on the welfare assessment. For example, if there is a change in size, number or bedding of the cubicles in a barn, there will be an almost immediate effect on the cleanliness and behaviour of the animals [32,33]. It would also have a direct effect on the housing scoring, as the protocol takes into account the softness and dimensions of the freestall. Therefore, housing is a good area to tackle if the farmer wants to see immediate results in the scoring. Instead, if a farmer vaccinates the animals for a certain disease, the results might not be immediate or significant due to multiple factors like disease status of the animals, virulence and prevalence of disease, etc. [34]. The same applies to feed and body condition score (BCS). A farmer can improve the diet of the animals, but if the health of the animals is not good, the results might not show up in the BCS of the animals [35]. 

It is no surprise that housing and health improvements had a positive interaction; since both contain many items related to hygiene, which is a key for having a high health status in any herd [36], and which is measured in the housing principle as the cleanliness of animals.

The fact that most types of improvement were related to an upgrade in their corresponding principle confirms that the improvements indeed ameliorate specific aspects of welfare and that the improvement categorization was done correctly. Feed came out differently. In this study, the feed principle was the principle with the most variation. This principle evaluates only two parameters: water supply and BCS. The BCS is a very limited parameter in the protocol as it takes only very lean cows into account [3]. In highly specialized farms as these ones, in order to increase profit, farmers tend to take special care for nutrition and have selected cows for high milk production. Highly producing animals tend to have lower body condition scores [37], something that the farmers cannot really modify with the in-farm recommendations they were given. Most advice for feed improvements was directed to feed and not water, and many of them would probably have a very indirect effect on the body condition score. For instance, improvements related to the control of mould in the silage likely affect more the health of the animals [38]. However, appropriate housing is essential for the cows to have enough lying time, which is directly correlated with the time spent in rumination [39,40]. Therefore, it is not strange that housing improvements increase the feed score since it might result in less extremely lean animals. Finally, the positive interaction between health and housing improvements with the feed principle confirms that the BCS could also be related to disease [28].

#### 3.6.4. New Categorizing

When the farms were re-categorized, for November 2013, 14 farms came out as enhanced-plus, 20 enhanced, four acceptable plus and 21 acceptable. For April 2015, 20 farms were enhanced-plus, 18 enhanced, three acceptable plus and 18 acceptable. Consequently, between both years, eight farms decreased their score by 1 or 2, 34 farms remained the same and 17 increased their score by 1, 2 or 3 categories. The number of improvements was significantly higher (by at least 10.75) in farms that had increased their category score by two (*p* = 0.004) or more. If the progress in category score was less than two, the difference in the number of improvements was not significant. In this case, health improvements were also the ones that had a significant effect on the change in category score (odds ratio 1:1.34, *p* = 0.016).

The new categorizing system divided most of the original categories into two. So if the difference for the new categorizing was only significant if it was by two score levels, there had not been any real progress regarding the number of improvements needed to change the category score. It was also not sensitive to more improvement types. Therefore, we conclude that changing the category score in this way was not very useful in dairy cattle, as it was in the study with broilers of Tuyttens et al. [20]. It was not expected that this change will encourage dairy farmers more to improve animal welfare in their farms.

#### 3.6.5. Economic Efficiency

Farms with a welfare category of ‘enhanced’ had, on average €189.82 more on their AMR than the farms of the ‘acceptable’ category (*p* = 0.01). Health score had also a positive impact on the economic result (OR 1:1.21; 95% CI 1.11, 1.31, *p* = 0.001). There was no significant relationship between the total sum score and the AMR (*p* = 0.12). 

The results suggest that the overall welfare of the animals could be positively related to production and even profit. This is not the first time that these kinds of relationships have been found. For example, Van Eerdenburg et al. [16] reported a positive correlation between their overall score for dairy cow comfort and milk yield. Many welfare aspects have been related in multiple studies to milk production, for example, BCS [37], stress and oxytocin [41], and water consumption [42]. Diseases have also been directly linked with production losses, for example, mastitis [43] and lameness [44,45]. This is probably the reason why health was the principle that had a significant effect on AMR. These results are quite promising since, as established before, the economic incentive is one of the best motivations for producers to work on animal welfare. Therefore, in further studies, the relationship of animal welfare with economic efficiency and profitability needs to be taken into account.

## 4. Conclusions

The newly developed Welfare Monitor is a practical instrument that takes about 1.5 h to execute on a farm with 100 cows. It consists of most measures of the WQ protocol after modifications and replacements by environment-based measures to make it faster to execute. It used the calculations and weights of WQ, except for three modifications, in order to make it more discriminative. The lengthy observation periods for the behavioural components were replaced by measures of the environment of the cows that were related to the behaviours. The result is a protocol that can be executed simply and quick, leaving the complex calculations to the computer (the full protocol is added in the Appendix A). 

Welfare assessment stimulated farmers to improve the conditions for their cows. The opinion of the farmers about the welfare assessment of their farm was not associated with the upgrade in score or improvements made over the two years but was most probably determined by the last scores obtained. It can be concluded that the recommendations made to the farmers actually helped improving animal welfare in their farms in multiple aspects. Investing in animal welfare will lead to better economic results because, as in other studies, the farms’ annual milk returns were positively related to animal welfare.

## Figures and Tables

**Figure 1 animals-11-00881-f001:**
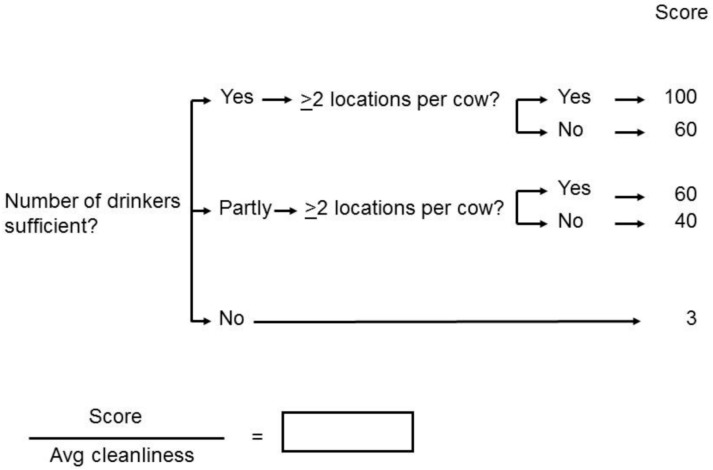
The scoring for water in the new Welfare Monitor for dairy cattle. For the determination of the number of drinkers being sufficient, the requirements of the Welfare Quality^®^ assessment protocol [3] (see Appendix A) were used. Then it was checked if there were at least two drinking locations available per cow [3]. The cleanliness was scored in points per drinker: clean = 1; partly dirty = 2; dirty = 3. The average of all drinkers was computed and used in the calculation: The result of the number and locations was divided by the average of the cleanliness.

**Figure 2 animals-11-00881-f002:**
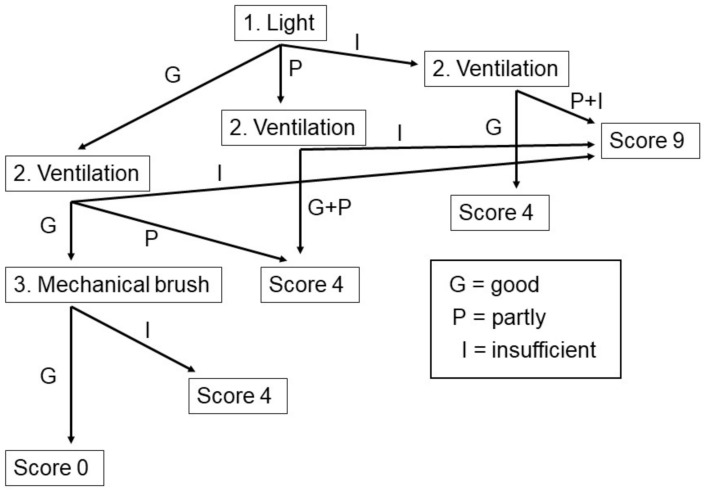
Flowchart for the scoring of the barn environment.

**Figure 3 animals-11-00881-f003:**
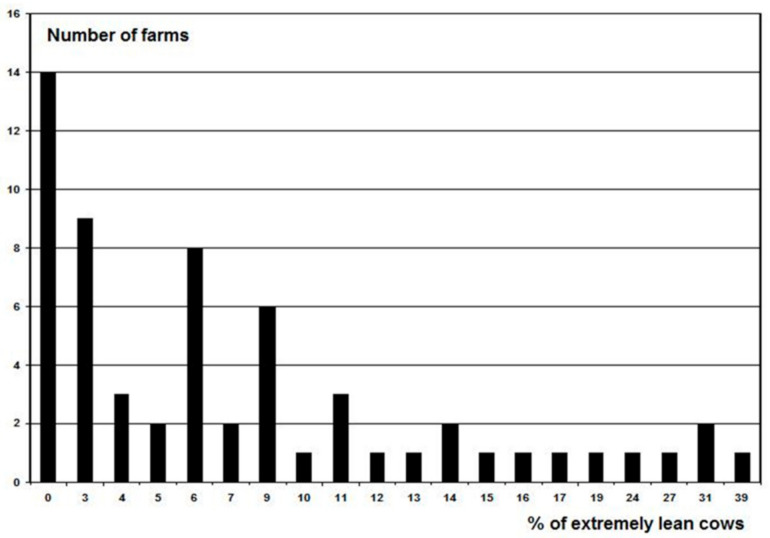
Frequency distribution of the percentage of extremely lean cows on the 60 farms.

**Figure 4 animals-11-00881-f004:**
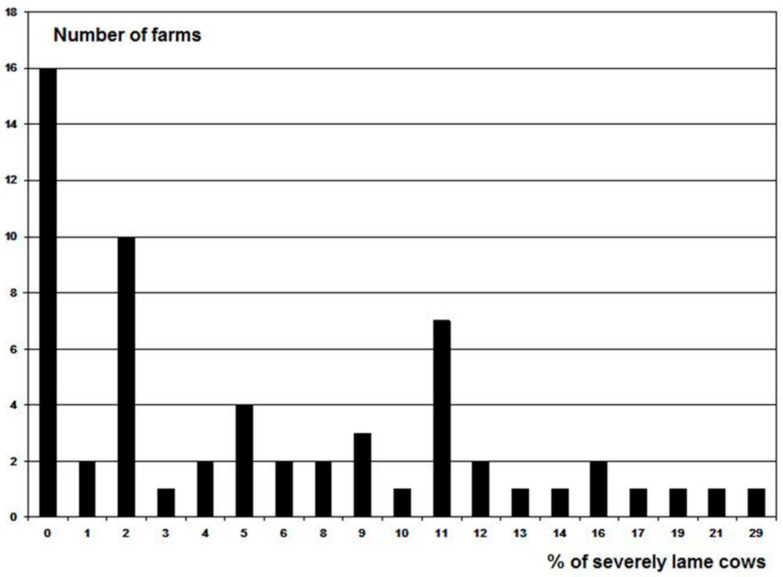
Frequency distribution of the percentage of severely lame cows on the 60 farms.

**Figure 5 animals-11-00881-f005:**
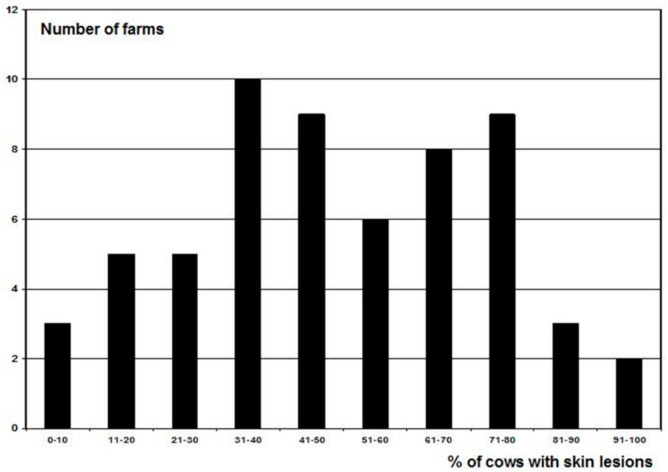
Frequency distribution of the percentage of cows with at least one skin lesion on the 60 farms.

**Table 1 animals-11-00881-t001:** Parameters measured in the new Welfare Monitor.

Principle	Parameters Measured
Feed and water	Body condition
	Water supply
Housing	Freestall dimensions
	Softness of bedding
	Cleanliness of the cows
	Access to pasture
	Cows lying outside the freestall
Health	Locomotion score
	Skin lesions
	Mastitis
	Other diseases (respiratory/metabolic/fertility)
Behaviour	Avoidance distance at the feeding fence
	Possibilities for expression of normal behaviour

**Table 2 animals-11-00881-t002:** Clinical scoring for the dirtiness of the skin.

Size of the Dirty Patch	1	2	3	4	5	Points
25 × 25–50 × 50 cm	>3	>2	>1	>0.5	≤0.5	
50 × 50 cm–one half Hind Quarter	>1.0	>0.5	>0.25	>0.15	≤0.15	
>one half Hind Quarter	>0.5	>0.25	>0.15	>0.1	≤0.1	

**Table 3 animals-11-00881-t003:** Minimal principle scores were necessary to achieve the welfare categories.

Current Categorizing System		New Categorizing System	
Excellent	>55 on all	Excellent Plus	>80 on all
	>80 on two	Excellent	>55 on all
			>80 on two
Enhanced	>20 on all	Enhanced Plus	>35 on all
			>55 on two
	>55 on two	Enhanced	>20 on all
			>50 on two
Acceptable	>10 on all	Acceptable plus	>10 on all
			>30 on three
	>20 on three	Acceptable	>10 on all
			>20 on three
Not classified		Not classified	

**Table 4 animals-11-00881-t004:** Correlations of the principle scores with the end classification of WQ and the new Welfare Monitor.

**Welfare Quality^®^ (WQ)**	***p***	**r**	**r^2^**
WQ Good Feeding	0.0001	0.515	0.265
WQ Good Housing	0.1448	0.190	0.036
WQ Good Health	0.7672	0.039	0.002
WQ Appropriate Behaviour	0.0061	0.349	0.122
**Welfare Monitor (WM)**	***p***	**r**	**r^2^**
WM Good Feeding	0.1078	0.274	0.075
WM Good Housing	0.0018	0.449	0.202
WM Good Health	0.0002	0.564	0.318
WM Appropriate Behaviour	0.0004	0.899	0.808

**Table 5 animals-11-00881-t005:** Average deviation in % of the original score for parameters in the WQ protocol when 75%, 66% or 50% of the animals was scored individually during the clinical inspection.

Items	75%	66%	50%
Lameness	5.5	6.8	9.9
Skin Lesions	5.2	6.1	9.3
Diseases	5.9	3.9	8.8
Health (principle)	4.9	4.5	7.9

**Table 6 animals-11-00881-t006:** Scores with the new Welfare Monitor of the farms in the years 2013 and 2015.

**Scores**	**April 2013**	**November 2013**	**April 2015**
	**# of Farms in Each Category Score**		
Excellent	0	0	0
Enhanced	6	34	38
Acceptable	30	25	21
Not classified	23	0	0
**Principle**	**Score Average (SD)**		
Feed	37.44 (17.94)	72.97 (21.18)	64.28 (20.97)
Housing	52.53 (7.98)	58.74 (5.91)	59.87 (5.26)
Health	43.20 (10.57)	39.20 (15.65)	42.40 (10.97)
Behaviour	26.42 (16.04)	40.21 (17.42)	40.03 (10.97)
Total score	159.59 (27.40)	211.11 (35.73)	206.51 (18.75)

**Table 7 animals-11-00881-t007:** The difference in classification between April 2013 and April 2015.

Difference	# of Farms That Changed Category
Decrease category by one	1
No change	15
Increase category by one	31
Increase category by two	12

**Table 8 animals-11-00881-t008:** Recommendations made to the farmers.

**Housing (18)**	**Health (19)**
Adjust the height of the feeding fence	Regularly remove waste from the silage pit
Incline feeding fence	Additional cleaning feed trough
More light in the shed	Improve mineral/vitamin supply
More cubicles (with respect to animals)	Better ventilation
Adjusting cubicle covering (litter)	Roughening slates
Deep litter cubicles	New slates (flat surfaces)
New/replace mattresses	More lime in cubicles
Move neck rail (diagonal length)	Treat scabies
Different (corrugated) neck rail	Vaccination mastitis
Adjust cubicle width	Vaccination (rest)
Adjust cubicle length	Selective dry-off
Give more headspace	Culling high somatic cell count cows
Brisket board (re)placement	Using barrier dip
Cleaning cubicles more frequently	Place flush system in the milking parlour
Cleaning slats more frequently	More active/earlier treatment claw problems
Purchased manure robot	Improve chemical mix footbath
Placed tube before feeding rack (no feed on slats)	More frequent/regular footbaths
Shave tails	More frequent/regular hoof trimming
	Cull severely lame cows
**Feed (14)**	**Behaviour (5)**
Prevent overheating and/or mould in silage	Rotating cow brush
Clean water bowls more frequently	Rubber on slates
Better silage covering (sand/tires)	Adjust breeding goals (behaviour)
Feeding speed increased (>1.5 m/week)	Calm treatment of cows
More feeding paces (with respect to animals)	Applying appropriate pasture system
Prevent overheating and/or mould in the feed bunk	
Concentrate incensement in early lactation	
Prevention of food selection: better mixed ration	
Individual feeding on condition	
Replace/repair (broken) water reservoirs	
Place additional water troughs	
Increase water pressure	
Ensure that rainwater does not run underneath the silage pit/better clearance of water	
Ensure that there is always enough feed available for cows/frequent shove feed	

**Table 9 animals-11-00881-t009:** Overview of the statistically significant results obtained in the statistical analysis.

Dependent Variable	Independent Variable	Effect	*p*-Value	Conclusion
Category score	Total improvements	Average difference: 7.44 improvements	0.007	The farms that changed positively in category score were the ones that made more improvements.
	Health improvements	Odds ratio: 1:1.5	0.014	If a farm made health improvements, it had a 0.5 higher chance of improving in category score.
Total sum score	Total improvements	OR 1:3.03; 95% CI 0.971, 9.47; β = 1.11	0.035	The number of improvements positively influenced the total scores.
	Housing improvements	OR 1.91; 95% CI 1.29, 2.81; β = 5.14	0.001	Working on housing improvements significantly increased the total sum score of the farms.
Principle scores	Housing improvements	Feed principle OR 1:1.46; 95% CI 1.084, 1.98; β = 3.09 Housing principle OR 1:2.47; 95% CI 1.54, 3.98; β = 0.905	0.044 0.001	The housing, health and behaviour principles are affected positively by its same type of improvement. The feed is also affected by housing improvements.
	Health improvements	Health principle OR 1:2.88; 95% CI 0.99, 8.29; β = 1.057	0.031	
	Behaviour improvements	Behaviour principle OR 1:1.84; 95% CI 1.06, 3.19; β = 3.24	0.015	
New category score	Total improvements	Mean difference between increasing in score by 0 and 2 = 10.75	0.004	Farmers that increased their category score by 2 in the new category scheme had made more improvements than those that did not increase their score.
	Health improvements	Odds ratio 1:1.34	0.016	If a health improvement was made, there was a 34% more probability of increasing their score.
Economic annual return (EAR)	Category score	Mean difference of €189.82	0.010	Farms with higher category score had a higher EAR
	Health score	OR 1:1.21; 95% CI 1.11, 1.31; β = 11.63	0.001	High health principle scores were positively related with a high EAR of the farm.

## Data Availability

The data are available in the Supplemental Materials

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
