# Peer review of "A New, Practical Animal Welfare Assessment for Dairy Farmers"

_animals, 2021, doi:10.3390/ani11030881_

Round 1
Reviewer 1 Report
The paper has been deeply improved and to my opinion are now suitable for publication.
Author Response
Thank you very much for your consideration.
Reviewer 2 Report
Thank you for resubmitting your revised manuscript; your study will be of interest to readers but I would suggest some further amendments are required prior to accepting for publication.
Line 20: suggest amending to: Besides promoting better welfare, this approach also led to a better financial result for the farms.
Line 275: please describe what descriptive analyses were conducted and ordinal logistic regression does not need to be capitalised
Line 323: please remove and at the start of the sentence
Table 4: please edit p=0.000 to p = 0.0004
Line 349: please add an s to principles
Line 447: what was measured for farmer opinion – need to state in method to be able to assess is analysed appropriately here and conclusions drawn are accurate
Line 447-448: you are stating personal opinion here, please justify via scientific debate
Line 450: this is your personal opinion, you could amend to anecdotally farmers in this study indicated a preference for a quick and straight forward approach
Table 9: seems to be repeated twice – amend to just include final version and should report beta, OR and CIs as well as p values for all variables included in this table
Line 493: suggest amending to relationship
In your results and discussion to consider the strength of correlations reported and for OLR would be normal to report OR and CIs in the main text of your results rather than the beta value as the OR define the relationship
Please also address discussion regarding the strength of your correlation coefficients as a number of these are weal and not meaningful for the measures – please refer to my previous feedback; if you wish to retain the current interpretation then a citation supporting your interpretation is required to justify the approach applied
Tables and figures: still feel there are too many tables and figures within the manuscript, suggest reconsidering if all are needed or if information could be presented in a different manner.
Supplementary files – excel data file: please remove all personal details from this file prior to publication
Author Response
R: Thank you for resubmitting your revised manuscript; your study will be of interest to readers but I would suggest some further amendments are required prior to accepting for publication.
>>> We thank the reviewer for the constructive comments.
R: Line 20: suggest amending to: Besides promoting better welfare, this approach also led to a better financial result for the farms.
>>> The sentence has been changed according to your suggestion.
R: Line 275: please describe what descriptive analyses were conducted and ordinal logistic regression does not need to be capitalized
>>> Done
R: Line 323: please remove and at the start of the sentence
>>> Done
R: Table 4: please edit p=0.000 to p = 0.0004
>>> Done
R: Line 349: please add an s to principles
>>> Done
R: Line 447: what was measured for farmer opinion – need to state in method to be able to assess is analysed appropriately here and conclusions drawn are accurate
>>> This is explained in line 240-242 of the M&M section.
R: Line 447-448: you are stating personal opinion here, please justify via scientific debate
>>> We used the word ‘probably’ here and we changed that into ‘possibly’. This is an indication that we do not have a scientifically based, but only an obvious explanation.
R: Line 450: this is your personal opinion, you could amend to anecdotally farmers in this study indicated a preference for a quick and straight forward approach
>>> Done
R: Table 9: seems to be repeated twice – amend to just include final version and should report beta, OR and CIs as well as p values for all variables included in this table
>>> The table is rather large, I tried to fit it on one page. To include all the requested numbers will make it even larger and some of those are not useful for the understanding here.
R: Line 493: suggest amending to relationship
>>> Done
R: In your results and discussion to consider the strength of correlations reported and for OLR would be normal to report OR and CIs in the main text of your results rather than the beta value as the OR define the relationship
>>> We added the OR and CI in both table 9 and the text, where applicable.
R: Please also address discussion regarding the strength of your correlation coefficients as a number of these are weal and not meaningful for the measures – please refer to my previous feedback; if you wish to retain the current interpretation then a citation supporting your interpretation is required to justify the approach applied.
>>> We did already mention in our discussion that there were many correlations that were weak (e.g. see line 447-448) The ones that we discuss in more detail are ones that have p values far below 0.05. The ones that are not meaningful are not discussed in detail for that reason.
R: Tables and figures: still feel there are too many tables and figures within the manuscript, suggest reconsidering if all are needed or if information could be presented in a different manner.
>>> I realize that there are many figures and tables, but this was a very large study with complex relationships. So I think that all tables and figures are needed.
R: Supplementary files – excel data file: please remove all personal details from this file prior to publication
>>> Ai !!! That was a serious mistake. The new file does not contain any names anymore
This manuscript is a resubmission of an earlier submission. The following is a list of the peer review reports and author responses from that submission.
Round 1
Reviewer 1 Report
The study is very interesting and there is a need for reliable protocols suitable to be routinely used by the farmers or veterinarians. The Authors make a great effort in the design but, as often happen in large studies with many parameters, the manuscript is not clear and hard to follow. It is a pity since the study has merit, in my opinion.
Therefore, I overall suggest improving the language used throughout the manuscript, to better describe the materials and methods and the results.
Introduction
The introduction is, in my opinion, well-centred, complete and provides all the information needed. Despite that, the text is a little cumbersome and I would suggest rewriting it to enhance the flow and improve the English language use.
The introduction stated that, instead of ABM, the protocol used resource-based measure. But on the table, there are not only RBM. To me, you made a selection of main indicators, which allows making the assessment rapidly without significant changes in the assessment, or with an improvement in the classification of the farms and on the suitability of the protocol by the farmers.
M&M
The content and methods are of interest. Anyway, I would suggest improving the exposition throughout this section. Most significant changes are listed below. Consider using diagrams or graph if needed.
Lines 103-115: This paragraph is interesting but hard to follow. Please rewrite it more logically and easily.
Lines 117-118: “The observers did not assess farms that they regularly visit and advise. One of the other observers assessed those farms. “ Please rewrite the sentence in a better way.
Line 124-126: The protocol is hard to follow. I would suggest to please better describe it.
Line 132-134: I have not understood what you did, please explain better.
Table 1: When you start to describe indicators, those are not all resource-based indicators, so please remove this statement throughout the text. You used a mix between ABM and resource-based the two of them.
Line 252: Please check the numeration of each section. Moreover, the statistical analysis only was referred to the opinion of the farmers, so at least it should be numbered as a subsection of the “farmer opinion” section (e.g. 2.2.1 ).
Line 267: Please change the title with something like: Definition of categories or simply Categorizing system.
RESULT and DISCUSSION
I can find everything that is needed in this section but it is quite difficult to actually find it. There are many information and results and more effort has to be put in the writing and organization of each section of the text. Please re-organize the text in a way easier to follow. And please revise the definition of animal-based measures and resource-based measures.
Here some example:
Line 322: Please start introducing shortly your paragraph. For example : “The correlation between principle scores with and the WQ protocol evidenced that Good feeding and Appropriate behaviour were the main principle influencing the classification.”
Line 345 and 348: Why there is two table description for tab 5?
Table 8: In my opinion, this table can be moved to supplementary files
CONCLUSION
The content of the conclusion is correct and allowed to properly resume the main findings of the study. Moreover, like the rest of the text, in my opinion, it needs to be improved in the language used.
Line 549: “except for 3” It sounds not clear to me.
Line 551: I don’t agree when you say “the direct environment”.
Reviewer 2 Report
An interesting study which is aiming to promote a viable and user-friendly alternative to the WQ for dairy farmers. There are a number of areas where clarification is needed to determine the key messages in the manuscript and I would advise the authors to consider how they interpret correlation coefficients and revisit inferences drawn based on the feedback provided. Streamlining content and focusing on key results and how these sit within the existing evidence base would enhance stricture and synthesis of the paper.
Simple summary:
Like the style of the summary and while I appreciate that the word count is limiting, I think it would be beneficial to change the final sentence re assessing farmer perception into a summary of the key results relating to your findings for this aspect.
Abstract:
Relevant summary provided
Line 20-21: suggest editing to remove reference to objective and make this more of an opening sentence; maybe state what WQ is, limitations and therefore what you did
Introduction:
Relevant background to study provided but scope to draw out rationale through evaluation of previous literature rather than this study
Line 45: could you provide a copy of the WQ as a supplementary file for readers less familiar with this measure? Or provide a URL to link to it
Line 47: not sure qualification is the correct word here, classification or categorisation may work better?
Line 54-59: I would avoid including the results of this study in your introduction and save this for the discussion, suggest removing and focusing on previous studies listed in next sentence to evaluate the WQ
Line 63 – 67: again here, this would sit in method as justification for your approach or results / discussion not in your intro, suggest removing
Line 86: is not a new para, first sentence is end of the preceding paragraph
Line 87-94: please remove reference to your study in line with comments above and focus on other work to introduce worth of analysis of farmer perception and move information related to your study to methods
Line 95-100: for me, this paragraph would also sit more readily in methods
Materials and methods
Formatting of paragraphs is not to journal guidelines
Line 106: please amend & to and; suggest changing mark to classification
Line 108-110: I agree better to use larger practices to reduce subjectivity but an individual vet will conduct the assessment so you will still get individual and subjective gradings for an individual farm unless multiple assessments – please clarify
Line 119: be careful here you can suggest it has a low discriminative capacity due to high number of farms graded as acceptable but they could be accurate gradings – please edit to reflect this e.g. amended to see if this was the case or not
Line 131-133: would be beneficial to include further detail re amended WM protocol as it is not fully clear (to me) how this was undertaken currently and need to make explicit so someone could repeat the WM method
Line 139, 142, 156, 222 : retain past tense, amend is to was
Line 155, 220: please amend are to were
Line 197-198: slightly odd presentation here, suggest editing to make it clearer need to do calculation to get P
Line 243: outline of questions, environment and participant demographics for focus groups should be provided
Line 244: please include copy of survey as a supplementary file and include details of distribution, content and analysis
Line 253: is opinion of the farmers a heading?
Statistical analysis: without full details of how farmer opinions were gathered it is difficult to follow this section
Descriptive analysis methods should also be outlined
Line 260: need to be careful here that your reader follows which tool is being judged to what – suggest editing this section to make this very explicit if comparing or testing within tools here
Line 267-274: further justification for this approach is required as changing categories could be misconstrued as manipulating data; highlight why greater specificity and sensitivity were needed and link to names WQ vs WM
Results and discussion
For this type of study, would be more usual to have two distinct sections for results and discussion
Line 286: suggest rewording to improve flow e.g. Farms scored XYZ under the original WQ protocol
Line 290: I agree there will be subjectivity but there is also the potential that no farm was excellent – avoid making assumptions (line 291) re selection without knowing this is due to subjectivity
Line 296: I am not that comfortable with an approach which matches observers best as this implies some degree of bias would be embedded in the study design; please clarify what point you are making here as I think currently it is difficult to follow (think you are stating in your study used multiple assessors which is a better approach? if yes then would like to see some reliability testing to analyse differences between assessors)
Line 299: please amend & to and
Line 302: suggest removing apparently as this is emotive and expanding debate as to why this percentage of lame cows is acceptable in WQ and whether this should be the case
Please ensure all figures are referred to in text and these would benefit from the inclusion of a legend to enable them to standalone.
Table 4: where p=0.000 this is SPSS rounding down ideally report as 0.0004
Line 332 and 336: your correlation coefficients in tables are weak and not meaningful for the measures (r<5 = by chance, 0.5-0.6 average – suggest including citation to outline interpretation of r), with the exception of WM appropriate behaviour which has a strong positive relationship with WM score, good health average positive relationship; please revisit results and discussion and reflect this
Line 362 and line 478: please edit sentence so it does not begin with and
Line 374-381: again here you are attributing meaningful relationships to correlations where the coefficient is not meaningful; please reconsider reporting and interpretation of these results
Some form of reliability analysis between observers should be integrated into assessment to demonstrate the reliability and validity of the tool.
Currently there are too many tables and figures within the manuscript, suggest reconsidering if all are needed or if information could be presented in a different manner.
Line 441: additional interpretation of Table 9 is required rather than stating this is a summary of statistical analysis; presentation of Table 9 is also quite confusing – suggest using a standardised format for all variables reported
Line 447: what was measured for farmer opinion – need to state in method to be able to assess is analysed appropriately here and conclusions drawn are accurate
Line 451-455: this is assumption as you do not know this is the reason for the differences suggest removing
Line 463: correlation coefficient should be reported as r not beta
Combining results and discussion has made some sections of the report quite difficult to follow and for the reader to judge if arguments presented are supported by results, research or supposition – would suggest the authors revisit these sections and streamline so key headline results are clearly presented and then discussed in context to aid flow, synthesis and clarity.
Limitations section is required.
Conclusions / animal welfare implications:
Animal welfare implications are not required for this journal, suggest moving these to discussion and keep conclusion focused to key take home messages
REVIEW: a new, practical animal welfare assessment for dairy farmers
An interesting study which is aiming to promote a viable and user-friendly alternative to the WQ for dairy farmers. There are a number of areas where clarification is needed to determine the key messages in the manuscript and I would advise the authors to consider how they interpret correlation coefficients and revisit inferences drawn based on the feedback provided. Streamlining content and focusing on key results and how these sit within the existing evidence base would enhance stricture and synthesis of the paper.
Simple summary:
Like the style of the summary and while I appreciate that the word count is limiting, I think it would be beneficial to change the final sentence re assessing farmer perception into a summary of the key results relating to your findings for this aspect.
Abstract:
Relevant summary provided
Line 20-21: suggest editing to remove reference to objective and make this more of an opening sentence; maybe state what WQ is, limitations and therefore what you did
Introduction:
Relevant background to study provided but scope to draw out rationale through evaluation of previous literature rather than this study
Line 45: could you provide a copy of the WQ as a supplementary file for readers less familiar with this measure? Or provide a URL to link to it
Line 47: not sure qualification is the correct word here, classification or categorisation may work better?
Line 54-59: I would avoid including the results of this study in your introduction and save this for the discussion, suggest removing and focusing on previous studies listed in next sentence to evaluate the WQ
Line 63 – 67: again here, this would sit in method as justification for your approach or results / discussion not in your intro, suggest removing
Line 86: is not a new para, first sentence is end of the preceding paragraph
Line 87-94: please remove reference to your study in line with comments above and focus on other work to introduce worth of analysis of farmer perception and move information related to your study to methods
Line 95-100: for me, this paragraph would also sit more readily in methods
Materials and methods
Formatting of paragraphs is not to journal guidelines
Line 106: please amend & to and; suggest changing mark to classification
Line 108-110: I agree better to use larger practices to reduce subjectivity but an individual vet will conduct the assessment so you will still get individual and subjective gradings for an individual farm unless multiple assessments – please clarify
Line 119: be careful here you can suggest it has a low discriminative capacity due to high number of farms graded as acceptable but they could be accurate gradings – please edit to reflect this e.g. amended to see if this was the case or not
Line 131-133: would be beneficial to include further detail re amended WM protocol as it is not fully clear (to me) how this was undertaken currently and need to make explicit so someone could repeat the WM method
Line 139, 142, 156, 222 : retain past tense, amend is to was
Line 155, 220: please amend are to were
Line 197-198: slightly odd presentation here, suggest editing to make it clearer need to do calculation to get P
Line 243: outline of questions, environment and participant demographics for focus groups should be provided
Line 244: please include copy of survey as a supplementary file and include details of distribution, content and analysis
Line 253: is opinion of the farmers a heading?
Statistical analysis: without full details of how farmer opinions were gathered it is difficult to follow this section
Descriptive analysis methods should also be outlined
Line 260: need to be careful here that your reader follows which tool is being judged to what – suggest editing this section to make this very explicit if comparing or testing within tools here
Line 267-274: further justification for this approach is required as changing categories could be misconstrued as manipulating data; highlight why greater specificity and sensitivity were needed and link to names WQ vs WM
Results and discussion
For this type of study, would be more usual to have two distinct sections for results and discussion
Line 286: suggest rewording to improve flow e.g. Farms scored XYZ under the original WQ protocol
Line 290: I agree there will be subjectivity but there is also the potential that no farm was excellent – avoid making assumptions (line 291) re selection without knowing this is due to subjectivity
Line 296: I am not that comfortable with an approach which matches observers best as this implies some degree of bias would be embedded in the study design; please clarify what point you are making here as I think currently it is difficult to follow (think you are stating in your study used multiple assessors which is a better approach? if yes then would like to see some reliability testing to analyse differences between assessors)
Line 299: please amend & to and
Line 302: suggest removing apparently as this is emotive and expanding debate as to why this percentage of lame cows is acceptable in WQ and whether this should be the case
Please ensure all figures are referred to in text and these would benefit from the inclusion of a legend to enable them to standalone.
Table 4: where p=0.000 this is SPSS rounding down ideally report as 0.0004
Line 332 and 336: your correlation coefficients in tables are weak and not meaningful for the measures (r<5 = by chance, 0.5-0.6 average – suggest including citation to outline interpretation of r), with the exception of WM appropriate behaviour which has a strong positive relationship with WM score, good health average positive relationship; please revisit results and discussion and reflect this
Line 362 and line 478: please edit sentence so it does not begin with and
Line 374-381: again here you are attributing meaningful relationships to correlations where the coefficient is not meaningful; please reconsider reporting and interpretation of these results
Some form of reliability analysis between observers should be integrated into assessment to demonstrate the reliability and validity of the tool.
Currently there are too many tables and figures within the manuscript, suggest reconsidering if all are needed or if information could be presented in a different manner.
Line 441: additional interpretation of Table 9 is required rather than stating this is a summary of statistical analysis; presentation of Table 9 is also quite confusing – suggest using a standardised format for all variables reported
Line 447: what was measured for farmer opinion – need to state in method to be able to assess is analysed appropriately here and conclusions drawn are accurate
Line 451-455: this is assumption as you do not know this is the reason for the differences suggest removing
Line 463: correlation coefficient should be reported as r not beta
Combining results and discussion has made some sections of the report quite difficult to follow and for the reader to judge if arguments presented are supported by results, research or supposition – would suggest the authors revisit these sections and streamline so key headline results are clearly presented and then discussed in context to aid flow, synthesis and clarity.
Limitations section is required.
Conclusions / animal welfare implications:
Animal welfare implications are not required for this journal, suggest moving these to discussion and keep conclusion focused to key take home messages
Reviewer 3 Report
This manuscript evaluated a new and practical animal welfare assessment for dairy farmers. I think that this is a very interesting topic. However, the article is very difficult to understand especially in the materials and methods section and it needs a complete rewriting. Also, the English language must be edited by a native speaker. Finally, there are several errors and bias in the manuscript and/or in the experimental design.
Introduction
Line 38: [.....] what does it mean?
Lines 54-59: This is no introduction, this is materials and methods.
Lines 59-67: This is no introduction, this is discussion.
Line 85: remove (Reasoned Action Approach;). You have already inserted the literature [19].
Lines 87-100: This is no introduction, this is materials and methods.
Materials and Methods
You did the WQ protocol on 60 different farms. The WQ protocol classified the farm with not classified, acceptable, enhanced, and excellent. How did you obtain the classification as good, average, or bad as reported in line 105?
For farm classification, you used several operators. Even if they had the same professional and experience level, an intra- and inter-operator coefficient of variation should be performed.
Lines 108-110: I did not agree. Also, this part concerning the classification of farms by different operators is not clear (i.e. it was a mean of each value evaluated by the vet?)
Lines 110-111: Again, how did you classify the farms in good, average and bad? Did you use the same farms of the WQ protocol? You have to use them to compare the WQ protocol with your new welfare monitor.
Lines 119-120: This is no materials and methods, this is results.
Lines 120-122: This is no materials and methods, this is discussion.
Lines 131-134: This is not clear, rewrite the sentence.
Figure 1: I think you have to change ≥ with ≤ after Partly.
Line 160: "If Diagonal < 185cm = 9 points; else if 185cm < Diagonal < 195cm = 4 points; else = 0 points". If you have 185 cm is 9 or 4 point? Insert ≥ or ≤.
Line 161: same as line 160, Insert ≥ or ≤.
Lines 163-164: same as line 160, Insert ≥ or ≤.
Line 165: Why did you multiple the 3 scores with 3?
Line169: Put H inside ().
Lines 169-170: same as line 160, Insert ≥ or ≤. If you have a score of 9 you obtain 0 or 4 points?
Lines 173-177: The example is wrong. "Example: 1.5% of the cows had a dirty patch size 25x25 – 50 x 50 cm; 0.6% had a dirty patch 50 x 50 cm – ½ hind quarter and 0.3% was dirty > ½ hind quarter. This will result in 3 + 1 + 2 = 6 points. The score for hygiene H=6 and B=9."
1.5% of the cows had a dirty patch size 25x25 – 50 x 50 cm = 3 points (>1 in the table), 0.6% had a dirty patch 50 x 50 cm – ½ hind quarter = 2 points (>0.5 in the table), 0.3% was dirty > ½ hind quarter = 2 points (>0.25 in the table). The score for hygiene is H=7 and B= 4 o 9 points????
Lines 213-214: Insert ≥ or ≤.
Line 241 "The farms were visited twice a year (April and November) for 2 years from 2013 to 2015". 2013, 2014, 2015 are 3 years, not 2.
Line 252: You pass from 2.2 in line 240 to 2.6 in line 252. Change in 2.3
Lines 253-254: Is not clear. Rewrite the sentence.
Line 260: 2.4 instead of 2.7
Line 267: 2.5 instead of 2.8
Line 278: 2.6 instead of 2.8
Results and discussion
The classification of the two different methods is not clear in materials and methods so for me is very difficult to understand the results.
Lines 287-288: With which protocol did you classify the farms?
Line 303: insert score after body condition.
Lines 341-343: I think that the deviations are high, especially for 50%.
Line 366: What does 40.16 mean?
Line 388: You have already classified hairless patches in line 204 as (HPs).
Lines 502-504: I did not agree with this sentence. The farmers should know these problems and how to solve them.
Lines 513-515: if you add up all the classification you obtain 59 farms in November 2013 and in April 2015 instead of 60 farms.
References 22 and 23: the http is not found.